# Hydrolysis of Element (White) Phosphorus under the Action of Heterometallic Cubane-Type Cluster {Mo_3_PdS_4_}

**DOI:** 10.3390/molecules26030538

**Published:** 2021-01-21

**Authors:** Airat M. Kuchkaev, Nikita Y. Shmelev, Aidar M. Kuchkaev, Aleksandr V. Sukhov, Vasily M. Babaev, Khasan R. Khayarov, Artem L. Gushchin, Maxim N. Sokolov, Oleg G. Sinyashin, Dmitry G. Yakhvarov

**Affiliations:** 1Arbuzov Institute of Organic and Physical Chemistry, FRC Kazan Scientific Center of the Russian Academy of Sciences, Arbuzov Str. 8, 420088 Kazan, Russia; 95ka@bk.ru (A.M.K.); nikulek94@mail.ru (N.Y.S.); kuchkaev95@mail.ru (A.M.K.); alex.suhoff@rambler.ru (A.V.S.); babaev@iopc.ru (V.M.B.); oleg@iopc.ru (O.G.S.); 2Alexander Butlerov Institute of Chemistry, Kazan Federal University, Kremlyovskaya Str. 18, 420008 Kazan, Russia; khayarov.kh@gmail.com (K.R.K.); 3Nikolaev Institute of Inorganic Chemistry SB RAS Lavrentiev av. 3, 630090 Novosibirsk, Russia; caesar@niic.nsc.ru (M.N.S.); 4Department of Natural Sciences, Novosibirsk State University, Pirogov Str. 2, 630090 Novosibirsk, Russia

**Keywords:** white phosphorus, heterometallic cubane-type clusters, molybdenum, palladium, phosphorous acid, phosphine

## Abstract

Reaction of heterometallic cubane-type cluster complexes—[Mo_3_{Pd(dba)}S_4_Cl_3_(dbbpy)_3_]PF_6_, [Mo_3_{Pd(tu)}S_4_Cl_3_(dbbpy)_3_]Cl and [Mo_3_{Pd(dba)}S_4_(acac)_3_(py)_3_]PF_6_, where dba—dibenzylideneacetone, dbbpy—4,4′-di-*tert*-butyl-2,2′-bipyridine, tu—thiourea, acac—acetylacetonate, py—pyridine, with white phosphorus (P_4_) in the presence of water leads to the formation of phosphorous acid H_3_PO_3_ as the major product. The crucial role of the Pd atom in the cluster core {Mo_3_PdS_4_} has been established in the hydrolytic activation of P_4_ molecule. The main intermediate of the process, the cluster complex [Mo_3_{PdP(OH)_3_}S_4_Cl_3_(dbbpy)_3_]^+^ with coordinated P(OH)_3_ molecule and phosphine PH_3_, have been detected by ^31^P NMR spectroscopy in the reaction mixture.

## 1. Introduction

Both organic and inorganic phosphorus-containing compounds have become widespread agents for various industrial applications. Traditional methods for the preparation of phosphorus compounds involve oxidation and chlorination of the element (white) phosphorus (P_4_) and use the phosphorus chlorides as phosphorylating agents for the synthesis of various organophosphorus substrates. It should be noted that direct activation and transformation of P_4_ is a very harsh and risky process that involves toxic and hazardous reagents and waste, and which negatively impacts on the environment.

The oxidation of P_4_ in the presence of H_2_O usually leads to the formation of phosphoric acid H_3_PO_4_, while phosphorous acid H_3_PO_3_ is more interesting and important phosphorus precursor which can be used as a phosphorylating agent as it contains a functionally capable P-H bond. Moreover, the current methods for the preparation of H_3_PO_3_ leave much to desire and require the use of toxic and hazardous phosphorus trichloride producing a huge amount of corrosive gaseous hydrogen chloride and dangerous phosphorus chloro-derivatives.

We have previously demonstrated that palladium complexes can be efficiently used for the preparation of phosphorous acid H_3_PO_3_ directly from P_4_ by its mild hydrolysis in the coordination sphere of the metal [1,2]. However, the main limitation of these catalytic systems is the formation of catalytically inactive palladium phosphides and palladium black [3]. It is important to note, that ruthenium-mediated P_4_ hydrolysis was thoroughly studied in Florence in the scientific group of M. Peruzzini and P. Stoppioni. It was shown that the hydrolysis of [CpRu(PPh_3_)_2_(η^1^-P_4_)] (Cp = cyclopentadienyl) complex leads to the formation of phosphine PH_3_ and phosphorous acid H_3_PO_3_ [4,5]. Moreover, it has been established that the mechanism of this process may involve the formation of binuclear intermediate, when η^2^-P_4_unite is doubly coordinated to two {CpRu(PPh_3_)_2_} moieties, which is hydrolyzed with the formation of H_3_PO_3_ and previously unknown 1-hydroxytriphosphane (PH(OH)PHPH_2_) as the intermediate of the overall process [6]. Later this process was improved and the kinetic of P_4_ hydrolysis was investigated using ruthenium complexes with water-soluble phosphine ligands [7]. There is also a notable example of stabilization of phosphorous acid H_3_PO_3_ in its tautomeric form P(OH)_3_ on the Ru site [8]. Hence, further development of new methods of white phosphorus hydrolysis is of high interest.

Heterometallic cubane-type cluster complexes with {M_3_PdS_4_} (M = Mo, W) core that were first described by Hidai’s group in Japan [9,10], possess a number of attractive properties including unordinary reactivity and catalytic activity of the Pd site [11,12,13,14,15,16,17,18,19,20,21,22]. To cite the most recent example, {Mo_3_PdS_4_} cluster complexes react with fullerene C_60_ to form hybrid compounds containing a fullerene molecule coordinated to palladium in the cluster core [23]. The {M_3_PdS_4_} clusters catalyze allylation of aromatics [24,25,26] and nucleophilic addition to triple bonds [10,27]. However, until now there have been no examples of P_4_ molecule coordination and activation in the coordination sphere of the {Mo_3_PdS_4_} cluster core, despite reported ability of such clusters to stabilize the unstable species or less-favoured tautomers, such as As(OH)_3_, P(OH)_3_, PhP(OH)_2_, Ph_2_P(OH), HP(OH)_2_ through coordination to the palladium site, which indicates high affinity of the latter for pnictogens. [28,29]. From these earlier studies, we have assumed that {Mo_3_PdS_4_} complexes could be involved in the process of P_4_ activation and its transformation.

In this work we investigated the reactivity of heterometallic complexes [Mo_3_{Pd(dba)}S_4_Cl_3_(dbbpy)_3_]PF_6_(**1**), [Mo_3_{Pd(tu)}S_4_Cl_3_(dbbpy)_3_]Cl (**2**) and [Mo_3_{Pd(dba)}S_4_(acac)_3_(py)_3_]PF_6_ (**3**) towards element (white) phosphorus.

## 2. Results and Discussion

### 2.1. Synthesis and Characterization

The trinuclear cluster complexes with the {M_3_S_4_} (M = Mo, W) core are able to incorporate a range of transition metals in low oxidation state (from 0 to +2), affording heterometallic cubane-type derivatives {M_3_M′S_4_}, where M′ = Cu, Ni, Pd, Pt, etc. [11,30,31,32,33,34,35].

The common approach for synthesis of the {M_3_M’S_4_} clusters involves the reaction of a low-valent metal precursor with a {M_3_S_4_} trinuclear complex in a desired coordination environment [12]. The same synthetic approach has been applied in current work, where [Pd_2_(dba)_3_] × CHCl_3_ was used as the palladium source. The synthetic routes to new cluster complexes **1** and **3** used in this work are depicted in Scheme 1. The cluster complex **2** has been obtained according to a previously published procedure [36].

The reaction of [Mo_3_S_4_Cl_3_(dbbpy)_3_]PF_6_ with [Pd_2_(dba)_3_] × CHCl_3_ (2:1 molar ratio) in CH_2_Cl_2_ gives complex **1** in 54% yield. The complex **3** was prepared in a similar way in 47% yield from [Mo_3_S_4_(acac)_3_(py)_3_]PF_6_ as starting compound. The formation and identity of **1** and **3** were confirmed by microanalysis and spectroscopic data.

The IR spectrum of **1** demonstrates the characteristic composite vibration bands, ν(C–C) and ν(C–N) at 1610 cm^–1^; ν(C-C), δ(C-H), δ(C-N) and δ(Mo-N) in the 1480–1410 cm^−1^ range; γ(C-H) and δ(C-H) in the 870–830 cm^−1^ and 1380–1310 cm^−1^ regions, and ring breathing bands from the coordinated dbbpy ligands in the 1028–906 and 766–423 cm^–1^ regions. The spectral data are in agreement with those reported for complex **2 [36]**. The intensive band (ν(C=O)) at 1625 cm^−1^ testifies the coordination of dba ligand to the palladium atom, while bands at 839 and 555 cm^−1^ belong to the hexafluorophosphate anion.

In the case of **3**, the IR spectrum is complicated by a significant overlap between bands from acac and py ligands. The band at 1603 cm^−1^ (ν(C=C + C-N)) relates to the bonded pyridine, and the two characteristic bands (ν(C=C + C=O)) at 1578 and 1523 cm^−1^ are associated with the acetylacetonate ligand. The two intensive bands at 838 and 556 cm^−1^ are explained by the presence of PF_6_^−^ group. The highly intensive band at 1627 cm^−1^ reveals the presence of coordinated dba [37].

The ^1^H NMR spectra of both **1** and **3** in CDCl_3_ demonstrate a complicated pattern due to the presence of overlapping signals in the 7–9 ppm area, that originate from the protons of various aromatic rings present in dbbpy, py, and dba ligands. There are also signals related to alkene fragments in dba (δ ~ 7 ppm). The characteristic signals generated by protons of the *tert*-butyl group (δ 1.40–1.45 ppm) in dbbpy ligand are detected in the case of **1**. The spectrum of **3** contains peaks associated with CH_3_- (δ 1.85 ppm) and CH-groups (δ 5.36 ppm) in acac.

The ESI-MS (+) of **1** contains the peaks with m/z ratio = 1669 ([Mo_3_{Pd(dba)}S_4_Cl_3_(dbbpy)_3_]^+^) (pseudomolecular peak), 1436 ([Mo_3_PdS_4_Cl_3_(dbbpy)_3_]^+^), 1476 ([Mo_3_{Pd(CH_3_CN)}S_4_Cl_3_(dbbpy)_3_]^+^), 1327 ([Mo_3_S_4_Cl_3_(dbbpy)_3_]^+^) which result from [Mo_3_{Pd(dba)}S_4_Cl_3_(dbbpy)_3_]^+^ fragmentation under experimental conditions. The signals with m/z ratio of 1057.8 ([Mo_3_PdS_4_(acac)_3_(py)_3_]^+^), 978.6 ([Mo_3_PdS_4_(acac)_3_(py)_2_]^+^), 899.6 ([Mo_3_PdS_4_(acac)_3_(py)]^+^), 951.8 ([Mo_3_S_4_(acac)_3_(py)_3_]^+^), 870.8 ([Mo_3_S_4_(acac)_3_(py)_2_]^+^), 793.8 ([Mo_3_S_4_(acac)_3_(py)]^+^), 714.7 ([Mo_3_S_4_(acac)_3_]^+^) are detected in the spectrum of **3**. The de-coordination of the pyridine ligands is expectable under ionization conditions and was also observed for the trinuclear {Mo_3_S_4_} precursor [38]. All the peaks have been assigned both from m/z and characteristic isotope patterns.

### 2.2. Interaction with P_4_

The reactivity of heterometallic cluster complexes [Mo_3_{Pd(dba)}S_4_Cl_3_(dbbpy)_3_]PF_6_ (**1**), [Mo_3_{Pd(tu)}S_4_Cl_3_(dbbpy)_3_]Cl (**2**) and [Mo_3_{Pd(dba)}S_4_(acac)_3_(py)_3_]PF_6_ (**3**) towards P_4_ was investigated both in the absence and in the presence of water. According to ^31^P NMR spectra, addition of the equimolar amount of P_4_ to the solutions of these complexes in DMF, THF and CH_2_Cl_2_ does not lead to transformation of P_4_ molecule, and no new signals from phosphorus-containing species were detected.

However, addition of water to the DMF solutions containing complexes **1** or **2** and P_4_ has led to the appearance of the signals associated with inorganic oxo-acids H_3_PO_3_ (δ 2.1 ppm) and H_3_PO_4_ (δ 0.9 ppm) with integral ratio of 3.4:1.0 (for **1**) and 7.5:1.0 (for **2**). Additionally, new signals around 110 ppm were detected. These signals correspond to the formation of [Mo_3_{PdP(OH)_3_}S_4_Cl_3_(dbbpy)_3_]^+^ (Figure 1), in which the Pd atoms bears the tautomeric form of phosphorous acid H_3_PO_3_ (P(OH)_3_) formed by the hydrolysis of white phosphorus. These results nicely fit with the previously published data [28,29], where the signals around 115 ppm in ^31^P NMR spectra were attributed to the complexes [Mo_3_{PdP(OH)_3_}S_4_(H_2_O)_9-x_Cl_x_]^(4–x)+^ obtained by the reaction of [Mo_3_{PdCl}S_4_(H_2_O)_9_]^3+^ with PCl_3_ or H_3_PO_3_ in 4M HCl. The total conversion of white phosphorus (by ^31^P NMR spectroscopy) in these reactions was 96.7% for complex **1** and 68.0% for complex **2**.

In order to boost the activity of clusters **1** and **2**, we attempted modification of their ligand surrounding by the substitution of the chloride-ions from the first and the second coordination spheres with weakly coordinated ions, and in this way to increase the electrophilic properties of Pd. Indeed, addition of TlNO_3_ as a halide scavenger increased the reactivity of the cluster towards P_4_. As a result, increased intensity of the signals related to H_3_PO_3_ and H_3_PO_4_, and the decreased intensity of the P_4_ signal were observed in ^31^P NMR spectra. In the case of complex **1**, full conversion of P_4_ was accomplished with a 72.0% yield of H_3_PO_3_. Complex **2** gave 74.4% conversion of P_4_ and 53.1% yield of H_3_PO_3_.

Addition of H_2_O to the reaction mixture containing [Mo_3_{Pd(dba)}S_4_(acac)_3_(py)_3_]PF_6_ (**3**) in DMF and P_4_ also allowed for the appearance of the signals associated with H_3_PO_3_ and H_3_PO_4_. The conversion of P_4_ (34.4%) was, however, substantially lower than with **1** and **2** and no signal attributed to [Mo_3_{PdP(OH)_3_}S_4_(acac)_3_(py)_3_]^+^ was observed in ^31^P NMR spectrum. This can be explained by assuming [Mo_3_{PdP(OH)_3_}S_4_(acac)_3_(py)_3_]^+^ being less stable than [Mo_3_{PdP(OH)_3_}S_4_Cl_3_(dbbpy)_3_]^+^, and quickly releases P(OH)_3_, which tautomerizes into the final product—phosphorous acid. The observed yield of H_3_PO_3_ was only 20.1%.

The solvent influence on the reactivity of P_4_ and its hydrolysis was also investigated, using the complex **1** as the benchmark. Addition of an excess of H_2_O to the reaction mixture containing **1** and P_4_ in CH_2_Cl_2_ allowed for detection of the signals associated with [Mo_3_{PdP(OH)_3_}S_4_Cl_3_(dbbpy)_3_]^+^ (δ 110 ppm), H_3_PO_3_ (δ 4.7 ppm) and H_3_PO_4_ (δ 1.1 ppm). Adding TlNO_3_ increased the signal intensities, and the observed molar ratio of H_3_PO_3_:H_3_PO_4_ was 3:2. The signal at δ −243.6 ppm related to the formation of PH_3_ was also observed, as minor peak. The conversion of P_4_ was 43.1% with only 8.6% yield of H_3_PO_3_.

In case of THF as the solvent, the addition of H_2_O to the reaction mixture containing **1** and P_4_ yielded the signal associated with [Mo_3_{PdP(OH)_3_}S_4_Cl_3_(dbbpy)_3_]^+^ with δ +113.8 ppm in ^31^P NMR spectrum. It is worth noting that there were no signals of any phosphorus-containing acids in ^31^P NMR spectra in this case. However, the activation of the complex with TlNO_3_ caused both the signal growth and the appearance of new signals related to H_3_PO_3_ (δ 3.3 ppm), H_3_PO_4_ (δ 1.1 ppm), and PH_3_ (δ −244.4 ppm). The integrated intensity ratio H_3_PO_3_:H_3_PO_4_ was 8:3, and the observed conversion of P_4_ was 85.5% with 24.4% yield of H_3_PO_3_.

The summary of the results obtained in the reaction of {Mo_3_PdS_4_} complexes with P_4_ is presented in Table 1.

It should be noted, that the presence of the Pd site in the cluster moiety to realise the hydrolysis of P_4_ molecule is mandatory, as this reaction does not proceed with Pd-free trinuclear cluster complexes [Mo_3_S_4_Cl_3_(dbbpy)_3_]Cl and [Mo_3_S_4_Cl_3_(dbbpy)_3_]PF_6_. This fact confirms that the transformation and followed hydrolysis of P_4_ requires the presence of Pd center. Moreover, the use of cluster core {Mo_3_PdS_4_} for hydrolysis of white phosphorus tetrahedron and its transformation into phosphorous acid (H_3_PO_3_) allows to avoid the formation of the insoluble and inactive Pd-black that is very important for the further use of these {Mo_3_PdS_4_} clusters as catalysts for the hydrolysis of white phosphorus.

## 3. Conclusions

Based on the experimental data, we can conclude that heterometallic cubane-type clusters [Mo_3_{Pd(dba)}S_4_Cl_3_(dbbpy)_3_]PF_6_ (**1**), [Mo_3_{Pd(tu)}S_4_Cl_3_(dbbpy)_3_]Cl (**2**) and [Mo_3_{Pd(dba)}S_4_(acac)_3_(py)_3_]PF_6_ (**3**) efficiently promote the hydrolysis of P_4_ molecule leading to the formation of H_3_PO_3_ as the major product. The complexes **1** and **2** bearing dbbpy ligand demonstrate higher activity in comparison with complex **3** containing acac ligand. Moreover, removal chloride anions from the coordination sphere of the cluster core with TlNO_3_ increases both the activity of the cluster complexes in P_4_ activation process and the yield of H_3_PO_3_. The use of the cluster with embedded Pd atom allows to avoid the Pd black formation which occurs when non-cluster Pd complexes are used for white phosphorus hydrolysis process. Thus, this work opens up prospects for studying the potential of heterometallic cubane-type clusters as catalysts for the selective conversion of white phosphorus to phosphorous acid. Further studies are in progress.

## 4. Experimental Section

CAUTION: White phosphorus and phosphine mentioned in this communication are hazardous compounds. White phosphorus needs to be stored under water in a well-ventilated dark place. White phosphorus is highly toxic and burns spontaneously when exposed to air. In an emergency, white phosphorus can be treated with aqueous copper(II) sulfate solution or sand. On contact with skin, white phosphorus gives highly painful, badly healing burns. In case of skin burns, washing with diluted aqueous solutions of KMnO_4_ or CuSO_4_ is advised. The continuous inhaling of white phosphorus vapors results in disease of the bone tissue, loss of teeth, and necrosis of parts of the jaw. An aqueous copper(II) sulfate solution (2%) can be used as an immediate antidote for poisoning. All reactions and handling of phosphine and white phosphorus must be carried out under an inert atmosphere in a well-ventilated hood.

All experiments related to the synthesis of the complexes, preparation of the solutions, solvents, and the manipulations with white phosphorus and all chemical reagents were performed under nitrogen atmosphere using standard Schlenk-line techniques.

Trinuclear precursors [Mo_3_S_4_Cl_3_(dbbpy)_3_]X (X = Cl^−^, PF_6_^−^; dbbpy—4,4′-di-*tert*-Bu-2,2′-bipyridine) [39] and [Mo_3_S_4_(acac)_3_(py)_3_]PF_6_ (acac—acetylacetonate, py—pyridine) [40] used to synthesize complexes **1**, **2** and **3** as well as complex [Mo_3_{Pd(tu)}S_4_Cl_3_(dbbpy)_3_]Cl (**2**) (tu—thiourea) [36], were prepared according to the published procedures. Commercially available reagents [Pd_2_(dba)_3_] × CHCl_3_ (dba—dibenzylideneacetone) (Sigma-Aldrich, Steiheim, Germany), thallium(I) nitrate (99.5%, ACROS Organics, Geel, Belgium) were used as purchased. White phosphorus was stored under a protective nitrogen atmosphere in a flask filled with water in a dark place and was washed sequentially in ethanol, acetone, and diethyl ether prior to use. 0.05 M THF solution was prepared by dissolving P_4_ in a required amount of solvent. Organic solvents were distilled before used.

Elemental C, H, N analyses were performed with a EuroEA3000 Eurovector analyzer (Eurovector SpA, Milano, Italy). IR spectraof samples in KBr pellets were recorded in the 4000–400 cm^−1^ range with a Perkin-Elmer System 2000 FTIR spectrometer(PerkinElmer, Waltham, Massachusetts, USA). ^1^H and ^31^P{^1^H} NMR spectra were registered at room temperature on a Bruker Avance III 400 MHz spectrometer (Bruker, New York, New York, USA)at frequencies of 400.0 (^1^H) and 161.9 (^31^P) MHz. UV–vis spectra were recorded with a Specord M40 (Carl Zeiss, Jena, Germany), Helios γ spectrophotometer (ThermoFisher Scientific, Waltham, Massachusetts, USA) in the 200–900 nm range in the CH_3_CN solution. A mass spectrometer (Agilent, 6130 Quadrupole MS, 1260 infinity LC, Santa Clara, California, USA) was utilized for the ESI measurements of **1**. The drying gas was nitrogen at a 300 L × h^−1^ flow rate. The sample solution (approx. 5 × 10^−5^ M) in acetonitrile was infused through a syringe pump directly into the interface at a flow rate of 0.4 mL min^−1^. The temperature of the source block was set to 120 °C and the interface to 150 °C. A capillary voltage of 2.0 kV was used in the positive scan mode, and low values of the cone voltage (Uc = 5–10 V) were used to control the extent of fragmentation. The observed isotopic pattern of each compound perfectly matched the theoretical isotope pattern calculated from their elemental composition by using the MassLynx 4.1 program (Waters Corporation, Milford, Massachusetts, USA). ESI measurements of **3** were performed using an AmazonX (Bruker Daltonics, Bremen, Germany) ion trap mass spectrometer in positive mode in the mass range of 200–3000 Da. The sample solution (approx. 5 × 10^−5^ M) in DMF was infused through a syringe pump directly into the interface at a flow rate of 0.2 mL × min^−1^. The ESI-MS conditions were as follows: capillary voltage, 2.5 kV; nitrogen drying gas, 10 L × min^−1^, 250 °C. Data processing was performed by DataAnalysis software (Bruker Daltonik GmbH, Version 4.0 SP4, Bremen, Germany).

Synthesis of [Mo_3_{Pd(dba)}S_4_Cl_3_(dbbpy)_3_]PF_6_ (**1**). A mixture of [Mo_3_S_4_Cl_3_(dbbpy)_3_]PF_6_ (0.2 g, 0.13 mmol) and [Pd_2_(dba)_3_] × CHCl_3_ (0.070 g, 0.068 mmol) was stirred in CH_2_Cl_2_ (20 mL) for 24 h. An excess of *n*-hexane was layered onto the resulting brown solution to give a brown product of **1** that was washed by *n*-hexane and diethyl ether. Yield: 0.133 g (54%).

IR (ν, cm^−1^): 3395 w, 3158 m, 3123 m, 2960 s, 2903 s, 2869 s, 1626 s 1618 s, 1549 m, 1477 m, 1464 m, 1415 s, 1311 w, 1295 w, 1254 m, 1201 w, 1158 w, 1129 w, 1020 m, 985 s, 975 w, 903 m, 885 w, 859 s, 830 m, 741 w, 720 w, 604 w, 555 s, 485 w, 427 w.

NMR ^1^H (400 MHz, CDCl_3_, 293 K): δ = 9.93 (d, 3H), 9.06 (3H), 8.49 (d, 3H), 8.31 (d, 3H), 7.80 (2H, dba), 7.28–7.65 (10H, dba), 7.31–7.39 (m, 6H, dbbpy), 7.01 (2H, dba), 1.45 (s, 27H, ^t^Bu), 1.40 (s, 27H, ^t^Bu) ppm.

ESI-MSI (+, CH_3_CN): m/z = 1671 ([Mo_3_{Pd(dba)}S_4_Cl_3_(dbbpy)_3_]^+^), 1437 ([Mo_3_PdS_4_Cl_3_(dbbpy)_3_]^+^), 1331 ([Mo_3_S_4_Cl_3_(dbbpy)_3_]^+^), 1313 ([Mo_3_S_4_Cl_2_(OH)(dbbpy)_3_]^+^), 1383 ([Mo_3_PdS_4_(OH)_3_(dbbpy)_3_]^+^), 1460 ([Mo_3_{Pd(CH_3_CN)}S_4_Cl_2_(OH)(dbbpy)_3_]^+^), 1277 ([Mo_3_S_4_(OH)_3_(dbbpy)_3_]^+^).

Anal. Calc. for C_71_H_86_Cl_3_F_6_Mo_3_N_6_OPPdS_4_: C, 47.0; H, 4.8; N, 4.6%. Found: C, 46.7; H, 4.4; N, 5.0%.

Synthesis of [Mo_3_{Pd(tu)}S_4_Cl_3_(dbbpy)_3_]Cl (**2**).A mixture of [Mo_3_S_4_Cl_3_(dbbpy)_3_]Cl (0.1 g, 0.74 mmol), [Pd_2_(dba)_3_] × CHCl_3_ (0.038 g, 0.37 mmol), and thiourea (0.056 g, 0.74 mmol) in 20 mL of CH_2_Cl_2_ was stirred for 24 h. An excess of hexane was layered onto the resulting brown solution to give a greenish-brown product of **2**. Yield: 0.080 g (71%). The obtained analytical data nicely fit with previously published results [36].

Synthesis of [Mo_3_{Pd(dba)}S_4_(acac)_3_(py)_3_]PF_6_ (**3**). A mixture of [Mo_3_S_4_(acac)_3_(py)_3_]PF_6_ (0.2 g, 0.17 mmol) and [Pd_2_(dba)_3_] × CHCl_3_ (0.089 g, 0.086 mmol) were stirred in CH_2_Cl_2_ (20 mL) for 24 h. An excess of *n*-hexane was layered onto the resulting brown solution to give a green-brown product of **3** that was washed by *n*-hexane and diethyl ether. Yield: 0.121 g (47%).

IR (ν, cm^−1^):2918 m, 1627s, 1611m, 1603 m, 1597s, 1578 m, 1568 m, 1523 s, 1485 m, 1444 s, 1416 m, 1366 s, 1280 s, 1177 w, 1151 w, 1066 s, 1021 s, 1013 s, 975 w, 876 s, 838 s, 782 w, 755 s, 698 s, 672 m, 622 m, 551 s, 540 m, 490s, 425 s.

NMR ^1^H (400 MHz, CDCl_3_, 293 K): δ = 9.58, 9.47 (6H, α-py), 8.15, 8.13, 8.10 (3H, γ-py), 7.80–7.30 (2H + 10H, dba; 6H, β-py), 7.03 (2H, dba) 5.36 (γ-CH, acac), 1.85 (CH_3_, acac) ppm.

ESI-MSI (+, DMF): m/z = 1057.8 ([Mo_3_PdS_4_(acac)_3_(py)_3_]^+^), 978.6 ([Mo_3_PdS_4_(acac)_3_(py)_2_]^+^), 899.6 ([Mo_3_PdS_4_(acac)_3_(py)]^+^), 951.8 ([Mo_3_S_4_(acac)_3_(py)_3_]^+^), 870.8 ([Mo_3_S_4_(acac)_3_(py)_2_]^+^), 793.8 ([Mo_3_S_4_(acac)_3_(py)]^+^), 714.7 ([Mo_3_S_4_(acac)_3_]^+^).

Anal. Calc. for C_47_H_50_F_6_Mo_3_N_3_O_7_PPdS_4_: C, 39.3; H, 3.5; N, 2.9%. Found: C, 39.0; H, 3.1; N, 3.3%.

Interaction of **1**, **2**, and **3** with P_4_ in the presence of H_2_O. A solution of P_4_ (0.0025 g, 0.02 mmol) in THF (0.4 mL) was added at room temperature to a solution of complex **1**, **2**, or **3** (0.02 mmol) in 1 mL of solvent (DMF, CH_2_Cl_2,_ or THF). Then, H_2_O (0.043 mL, 2.4 mmol) was added dropwise to the reaction mixture. After 12 h of stirring, TlNO_3_ (0.021 g, 0.08 mmol) was added. An analysis of the reaction mixture was provided by ^31^P NMR spectroscopy after each of the consequent steps.

## Data Availability

Data sharing is not applicable to this article.

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
