# Peer review of "Hydrolysis of Element (White) Phosphorus under the Action of Heterometallic Cubane-Type Cluster {Mo3PdS4}"

_molecules, 2021, doi:10.3390/molecules26030538_

Round 1

Reviewer 1 Report

Yakhvarov and coworkers describe in this interesting paper the application of some hetero-cubane clusters bearing the tetrametallic Mo3PdS4 core to the production of phosphorus oxyacids with particular emphasis to H3PO3, a largely sought intermediate for the industrial synthesis of organophosphites and other organophosphorus compounds. The chemistry here presented starts from the previously documented ability of this kind of cubanes to tautomerize H3PO3 to the trishydroxyphosphine form and expands this chemistry to include the reactivity of this class of cubanes towards white phosphorus. The latter molecule, in the presence of water, is firstly hydrolyzed to afford clusters containing the [Mo3{Pd{P(OH)3}S4] unit before releasing H3PO3 in the reaction medium. The role of the solvent and the nature of different ancillary ligands sitting atop the palladium and the molybdenum atoms is also briefly investigated. Although the reaction mechanism remains obscure and no mechanistic consideration is provided, the article reports an interesting piece of chemistry already in its present form and can be published in Molecules after a revision which takes into consideration the following points.

1) The English needs to be fixed up throughout the manuscript.

2) The parentheses coding is incorrect, particularly in the formula of the clusters used as reagents. I’d suggest to use also curly brackets when necessary. For example, for complex 3, the notation [Mo3{Pd(dba)}S4(acac)3(py)3]PF6 should be used. I have also noticed that there is some inhomogeneity in writing the formulas of the compounds. For example complex 3 is written as [Mo3(Pd(dba))S4(py)3(acac)3]PF6 in scheme 1 (under the sketch) while it is formulated as [Mo3(Pd(dba))S4(acac)3(py)3]PF6 in the text. Similar inaccuracies are spread in the manuscript and should be carefully revised.

3)There are important omissions in the literature listed in the reference section. In particular, the metal-mediated hydrolysis of P4 was documented more widely than mentioned in reference 2. The following papers report on the hydrolysis of P4 in the coordination sphere of a family of Cp2Ru(L) complexes, which produces inter alia H3PO3 via tautomerization of P(OH)3. The mechanism of the reaction is also discussed. The papers to be added to the list of references are: a) P. Barbaro et al. Angew. Chem. Int. Ed. 2008, 47, 4425; b) D. Akbayeva et al. Compt. Rend. Chim. 2010, 13, 395. Also relevant to the tautomerization of P(OH)3 to H3PO3 is the following article: D. Akbayeva et al. Dalton. Trans. 2006, 389.

4) In chapter 2.2 it is stated that the heterocubane clusters here investigated behave as effective catalysts in promoting the hydrolysis of P4 to yield phosphorous acid. The same statement is also highlighted in the conclusion section. However, in spite of these claims, inspection of the experimental section does not reveal any catalytic experiment as the reactions described in the last paragraph of the experimental part are strictly stoichiometric processes and no catalytic test is reported. I’d advise the authors to tune down these statements and remove any mention to a catalytic reaction, which, at the present stage of the study is a sort of flight forward in this chemistry.

Author Response

Dear Dr. Yan,

Thank you for your e-mail of January 13th, 2021 and for deciding that our manuscript ICA_2020_76 may be reconsidered for publication in Molecules taking into account the revisions proposed by Reviewers 1, 2 and 3. Details on the manuscript changes are given below. All changes included in the manuscript have been highlighted in yellow.

1) The English needs to be fixed up throughout the manuscript.

Answer: The text of the manuscript has been carefully proofread in relation to the English and the corresponding corrections were made.

2) The parentheses coding is incorrect, particularly in the formula of the clusters used as reagents. I’d suggest to use also curly brackets when necessary. For example, for complex 3, the notation [Mo3{Pd(dba)}S4(acac)3(py)3]PF6 should be used. I have also noticed that there is some inhomogeneity in writing the formulas of the compounds. For example complex is written as [Mo3(Pd(dba))S4(py)3(acac)3]PF6 in scheme 1 (under the sketch) while it is formulated as [Mo3(Pd(dba))S4(acac)3(py)3]PF6 in the text. Similar inaccuracies are spread in the manuscript and should be carefully revised.

Answer: All chemical formulas in the manuscript have been revised and presented now in accordance with recommendation and common standard.

3) There are important omissions in the literature listed in the reference section. In particular, the metal-mediated hydrolysis of P4 was documented more widely than mentioned in reference 2. The following papers report on the hydrolysis of P4 in the coordination sphere of a family of Cp2Ru(L) complexes, which produces inter alia H3PO3 via tautomerization of P(OH)3. The mechanism of the reaction is also discussed. The papers to be added to the list of references are: a) P. Barbaro et al. Angew. Chem. Int. Ed. 2008, 47, 4425; b) D. Akbayeva et al. Compt. Rend. Chim. 2010, 13, 395. Also relevant to the tautomerization of P(OH)3 to H3PO3 is the following article: D. Akbayeva et al. Dalton. Trans. 2006, 389.

Answer: We are very thankful to the Reviewer for this very useful remark that expand the scope of the used transition metals for the hydrolysis of P4! The mentioned by the Reviewer references have been added to the Reference list with the corresponding remarks in the main body text. Unfortunately, we were not able to find the mentioned paper D. Akbayeva et al. Comptes Rendus Chimie, 2010, 13, 395 which is probably related to the manuscript Di Vaira, M., Peruzzini, M., & Stoppioni, P. Comptes Rendus Chimie, 2010, 13 (8-9), 935-942 which is nicely fit with the points mentioned by the Reviewer. This latter manuscript also has been added to the Reference list of the paper.

4) In chapter 2.2 it is stated that the heterocubane clusters here investigated behave as effective catalysts in promoting the hydrolysis of P4 to yield phosphorous acid. The same statement is also highlighted in the conclusion section. However, in spite of these claims, inspection of the experimental section does not reveal any catalytic experiment as the reactions described in the last paragraph of the experimental part are strictly stoichiometric processes and no catalytic test is reported. I’d advise the authors to tune down these statements and remove any mention to a catalytic reaction, which, at the present stage of the study is a sort of flight forward in this chemistry.

Answer: Some catalytic experiments have already been carried out, but as the Reviewer correctly noted, these data are not presented in this manuscript. We are talking only about stoichiometric processes. We absolutely agree with the proposal of the Reviewer, and all mentions to catalysis were removed from the text.

We appreciate the comment of the Reviewers that have helped us to improve the present manuscript and hope that it will be acceptable in the corrected form.

Sincerely yours,

Artem Gushchin

Reviewer 2 Report

This communication reports the reactivity of heterometallic sulfide clusters toward white phosphorous (P4) in the presence of water to give H3PO3.  Crucial role of the cluster core {Mo3PdS4} is confirmed by comparing with the Pd-free clusters and their previous works using mono-Pd complexes.  The reactions were traced by 31P NMR method, which showed formation of H3PO3 as a main product as well as some species such as H3PO4, PH3, and the cluster-bound P(OH)3.  Conversions of P4 and yields H3PO3 were determined by NMR and summarized in Table 1 for each reactions.  However, only partial data are provided for the amounts of other products.  For example, the ratio of H3PO3 to H3PO4 for some reactions are described in text.  Adding these data in Table 1 will be helpful for readers to follow the fate of P atoms.  The stoichiometry of the reaction reported herein is quite complicated and does not seem to be a simple hydrolysis.  The predominantly formed H3PO3 is an oxidized product of elemental phosphorous, while not so much amount of the reduced products such as PH3 have been detected.  Rare examples of true hydrolysis of P4 have been reported for Ru and Os complexes by Stoppioni group (Dalton 2005, 2234; ACIE 2008, 47, 4425).  Although equimolar reactions (Mo3PdS4:P4 = 1:1) were examined in this study, unique effect of {Mo3PdS4} core is valuable and invokes expectation in catalytic application.  

Before publication, I hope that the authors give careful consideration to the following.

1) Microanalysis data are missing in Experimental section.  Caution to phosphorous materials is written very well.

2) For compound 1, 1H NMR signals of totally 16H intensity are assigned to the dba ligand (C17H14O).  The dbbpy signals are consistent with the proposed structure.

3) Stereo-chemistry of 3 shown in Scheme 1 should be carefully considered.  The structure of the trinuclear cluster precursor with acac ligands is drawn in a C3 symmetry, although it has been crystallographically confirmed to have a C3v symmetry (ref. 36).  The same molecular symmetry is suggested from the 1H NMR spectra which shows single CH3 signals of acac.

4) Some texts need correction of style: e.g. line 238, “0,1 g,” “0,038 g,” “0,056 g.” 

Author Response

Dear Dr. Yan,

Thank you for your e-mail of January 13th, 2021 and for deciding that our manuscript ICA_2020_76 may be reconsidered for publication in Molecules taking into account the revisions proposed by Reviewers 1, 2 and 3. Details on the manuscript changes are given below. All changes included in the manuscript have been highlighted in yellow.

 Conversions of P4 and yields H3PO3 were determined by NMR and summarized in Table 1 for each reactions.  However, only partial data are provided for the amounts of other products.  For example, the ratio of H3PO3 to H3PO4 for some reactions are described in text.  Adding these data in Table 1 will be helpful for readers to follow the fate of P atoms. 

Answer: Table 1 has been supplemented with data on phosphoric acid (H3PO4).

The stoichiometry of the reaction reported herein is quite complicated and does not seem to be a simple hydrolysis.  The predominantly formed H3PO3 is an oxidized product of elemental phosphorous, while not so much amount of the reduced products such as PH3 have been detected.  Rare examples of true hydrolysis of P4 have been reported for Ru and Os complexes by Stoppioni group (Dalton 2005, 2234; ACIE 2008, 47, 4425).

Answer: Thanks a lot for this remark! Indeed, the stoichiometry of the reaction is complicated and may involve several stages and intermediates. Unfortunately, it is very difficult to discuss regarding the mechanism of P4 hydrolysis in the coordination sphere of palladium. However, the mentioned by the Reviewer papers can disclose some analogy of this process based on Ru and Os complexes. Thus, we have cited the mentioned articles. It should be noted that coordination behaviour of the formed PH3 may be different for Pd and Ru (Os) complexes and probably it does not form stable complexes with {Mo3PdS4} core of the used clusters and can be easily removed from the reaction mixture in the gaseous form.

1) Microanalysis data are missing in Experimental section.  Caution to phosphorous materials is written very well.

Answer: Microanalysis has been provided and the relevant data were added to the manuscript.

2) For compound 11H NMR signals of totally 16H intensity are assigned to the dba ligand (C17H14O).  The dbbpy signals are consistent with the proposed structure.

Answer: The 1H NMR spectra has been reanalysed and the data of proton intensities related to the dba ligand (C17H14O) were corrected. 

3) Stereo-chemistry of 3 shown in Scheme 1 should be carefully considered.  The structure of the trinuclear cluster precursor with acac ligands is drawn in a C3 symmetry, although it has been crystallographically confirmed to have a C3v symmetry (ref. 36).  The same molecular symmetry is suggested from the 1H NMR spectra which shows single CH3 signals of acac.

Answer: Scheme 1 of has been revised. The structure of the trinuclear precursor [(Mo3S4(acac)3(py)3]+) has been redrawn in accordance with the established (ref. 40, new numbering) symmetry.

4) Some texts need correction of style: e.g. line 238, “0,1 g,” “0,038 g,” “0,056 g.” 

Answer: The style of digits has been revised and presented now in accordance with common standard.

We appreciate the comment of the Reviewers that have helped us to improve the present manuscript and hope that it will be acceptable in the corrected form.

Sincerely yours,

Artem Gushchin

Reviewer 3 Report

The manuscript from Gushchin and Yakhvarov describes the reactions between different Mo/Pd-S clusters and white phosphorus in water. The research is quite interesting and appears to have been well done. I’m struggling a little bit to understand exactly what the authors mean in terms of their yields and conversions (and if this is truly catalytic), so I would recommend major revisions according to the points below. At present there is a lack of clarity around what is actually being presented.

  1. I noticed the word ‘phosphorous’ (the name of the acid) being used to describe the element phosphorus. One example that jumps out is on page 2, line 60. There may be other examples in the paper – please check.
  2. P3, line112: Does ‘the equivalent amount…’ mean ‘one equivalent’?
  3. I would like the authors to add a figure depicting their reaction product – i.e. a PO3H3 ligand on the Pd.
  4. The authors describe the behaviour as catalytic – but isn’t this just stoichiometric? I think the authors should look to significantly clarify the amounts of the different species used. Can the authors write a balanced equation for the reaction of 1 with P4?
  5. Is all of the H3PO3 coordinated to the clusters? Can it be recovered? Again, I think this needs to be made far clearer in the paper.

Author Response

Dear Dr. Yan,

Thank you for your e-mail of January 13th, 2021 and for deciding that our manuscript ICA_2020_76 may be reconsidered for publication in Molecules taking into account the revisions proposed by Reviewers 1, 2 and 3. Details on the manuscript changes are given below. All changes included in the manuscript have been highlighted in yellow.

 1.I noticed the word ‘phosphorous’ (the name of the acid) being used to describe the element phosphorus. One example that jumps out is on page 2, line 60. There may be other examples in the paper – please check.

Answer: The spelling has been corrected throughout manuscript.

2. P3, line112: Does ‘the equivalent amount…’ mean ‘one equivalent’?

Answer: The word ‘equivalent’ has been changed with ‘equimolar’ that corresponds real conditions. 

3. I would like the authors to add a figure depicting their reaction product – i.e. a PO3H3 ligand on the Pd.

Answer: The Figure 1 depicting [Mo3{PdP(OH)3}S4Cl3(dbbpy)3]+ has been added to the manuscript.

4. The authors describe the behaviour as catalytic – but isn’t this just stoichiometric? I think the authors should look to significantly clarify the amounts of the different species used. Can the authors write a balanced equation for the reaction of 1 with P4?

Answer: This remark echoes that of Reviewer 1 (see response to Remark 4). To make it clearer, we have reformulated some of the proposals and removed the mention of catalysis.

5. Is all of the H3PO3 coordinated to the clusters? Can it be recovered? Again, I think this needs to be made for clearer in the paper.

Answer: Thank you for this remark. Fortunately, not all formed amount of H3PO3 is coordinated to the clusters. This allows to get H3PO3 in its free form without application of the decomplexation technique. However, when the concentration of H3PO3 is increased, the factor of the concentration makes more favourable the formation of the complexed form of H3PO3 (in its tautomeric form P(OH)3).

We appreciate the comment of the Reviewers that have helped us to improve the present manuscript and hope that it will be acceptable in the corrected form.

Sincerely yours,

Artem Gushchin

Round 2

Reviewer 3 Report

The authors have addressed all of my comments. I recommend that the manuscript is accepted for publication.